# Fostering Appropriate Antibiotic Use in a Complex Intervention: Mixed-Methods Process Evaluation Alongside the Cluster-Randomized Trial ARena

**DOI:** 10.3390/antibiotics9120878

**Published:** 2020-12-08

**Authors:** Regina Poss-Doering, Lukas Kühn, Martina Kamradt, Anna Stürmlinger, Katharina Glassen, Edith Andres, Petra Kaufmann-Kolle, Veit Wambach, Lutz Bader, Joachim Szecsenyi, Michel Wensing

**Affiliations:** 1Department of General Practice and Health Services Research, University Hospital Heidelberg, Im Neuenheimer Feld 130.3, 69120 Heidelberg, Germany; lukas.kuehn@med.uni-heidelberg.de (L.K.); martina.kamradt@med.uni-heidelberg.de (M.K.); anna.stuermlinger@me.com (A.S.); katharina.glassen@med.uni-heidelberg.de (K.G.); joachim.szecsenyi@med.uni-heidelberg.de (J.S.); michel.wensing@med.uni-heidelberg.de (M.W.); 2aQua Institut, Maschmuehlenweg 8-10, 37073 Goettingen, Germany; edith.andres@aqua-institut.de (E.A.); petra.kaufmann-kolle@aqua-institut.de (P.K.-K.); 3Agentur deutscher Arztnetze e.V., Friedrichstraße 171, 10117 Berlin, Germany; info@drwambach.de; 4Kassenärztliche Vereinigung Bayerns (KVB), 80684 München, Germany; baderlochham@gmail.com

**Keywords:** appropriate antibiotics use, primary care, quality improvement, mixed-methods

## Abstract

The cluster randomized trial ARena (sustainable reduction of antibiotic-induced antimicrobial resistance, 2017–2020) promoted appropriate use of antibiotics for acute non-complicated infections in primary care networks (PCNs) in Germany. A process evaluation assessed determinants of practice and explored factors associated with antibiotic prescribing patterns. This work describes its findings on uptake and impacts of the complex intervention program and indicates potential implementation into routine care. In a nested mixed-methods approach, a three-wave study-specific survey for participating physicians and medical assistants assessed potential impacts and uptake of the complex intervention program. Stakeholders received a one-time online questionnaire to reflect on network-related aspects. Semi-structured, open-ended interviews, with a purposive sample of physicians, medical assistants and stakeholders, explored program component acceptance for daily practice and perceived sustainability of intervention component effects. Intervention components were perceived to be smoothly integrable into practice routines. The highest uptake was reported for educational components: feedback reports, background information, e-learning modules and disease-specific quality circles (QCs). Participation in PCNs was seen as the motivational factor for guideline-oriented patient care and adoption of new routines. Future approaches to fostering appropriate antibiotics use by targeting health literacy competencies and clinician’s therapy decisions should combine evidence-based information sources, audit and feedback reports and QCs.

## 1. Introduction

Antibiotics are powerful medicines that can mitigate bacterial infections and save lives when used appropriately. Driven by a still widespread use of antibiotics, antimicrobial resistance remains a challenge for healthcare systems all over the world, leading to high costs, diminishing treatment options and increased mortality [1]. In the United States of America alone, more than 2.8 million antibiotic-resistant infections are reported per year with 35,900 resulting deaths [2]. Surveillance data for 30 European countries show that each year more than 670,000 antibiotic-resistant infections occur causing approximately 33,000 deaths [3,4]. By and large, a growing list of infections that have become harder to treat and emerging new resistance mechanisms have made changes in the way antibiotics are prescribed and used indispensable [1,2].

In Germany, where about 90% of the used antibiotics are prescribed in ambulatory care, measures have been launched to foster the appropriate antibiotics use and aim at a sustainable reduction of antibiotics misuse and overuse. In this context, a national strategy is being pursued [5,6] and a number of initiatives and scientific studies are carried out to strengthen the One Health approach, monitor the development of resistances, foster adequate competencies and to preserve existing treatment options [7].

The cluster randomized trial ARena (German: Antibiotika-Resistenzentwicklung nachhaltig abwenden; English: Sustainable reduction of antibiotic-induced antimicrobial resistance, 2017–2019, trial registration: ISRCTN, ISRCTN58150046) intended to promote the rational, appropriate use of antibiotics for acute non-complicated infections in primary care in Germany [8]. In a multifaceted strategy, ARena used multiple interacting intervention components to address primary care provider and patient knowledge and attitudes about the use of antibiotics [8]. Embedded into 14 primary care networks (PCNs) across the German federal states of Bavaria and North-Rhine Westphalia, the approach to foster appropriate antibiotics use was based on strengthening awareness and understanding of the challenge of antimicrobial resistances (AMR). PCNs support primary care practices regarding quality improvement, administration and reimbursement. Effective communication, education and training were addressed to PCNs physicians, their care teams and the regional public. To facilitate insights into determinants of appropriate antibiotics use and into influences and mechanisms of action, a process evaluation accompanied the complex intervention program in ARena to explore factors and processes leading to impacts on antibiotic prescribing patterns [8].

For complex interventions, process evaluations can be used to understand the functioning of the intervention by investigating uptake of intervention components, mechanisms of impact and contextual factors and to compliment high-quality outcome evaluations [9]. The process evaluation conducted alongside ARena aimed not only to assess whether program components were implemented as intended, and which were perceived to have impact, but also to thoroughly identify and analyze efforts for integration of components as well as determining organizational and individual factors of the appropriate antibiotics use within the ARena program. This present work aims to describe the identified individual and organizational factors affecting the uptake of this multi-faceted program and to offer indications referring to a potential implementation of tested components into routine care.

## 2. Results

### 2.1. Overview

Reporting of the findings of this mixed methods study follows the structure and domains of the applied analytical framework to integrate results from the survey and the interview study.

### 2.2. Sociodemographic Characteristics of Survey Respondents

Response rates for the survey were 75.6%, 66.2% and 63.3% for physicians and 93.0%, 83.9% and 68.2% for non-physician health professionals (T0, T1, T2) (comparable to medical assistants (MAs) in USA [10]). In T0, 229 physicians and 80 MAs returned the questionnaires, in T1 200 physicians and 73 MAs responded, in T2 184 physicians and 58 MAs. Gender distribution in physicians in T0 was at 34% of female respondents (32% in T2). In comparison, the gender distribution of MAs was 100% female over all measurement points. Physicians had a mean age of 54 years, which did not change over time. MAs were 39 years of age, which did not vary over measurement points. In the online survey, 10 PCN management representatives responded (71.4%). Further information about sociodemographic characteristics of survey respondents is displayed in Table 1. Sociodemographic data were reported in T0 and T2 questionnaires only.

### 2.3. Sociodemographic Characteristics of Interview Participants

In the first phase of qualitative data collection in 2018, 45 interviews were conducted. Of these, 27 interviews were carried out with physicians, 11 with MAs and seven with stakeholders. The mean age of participants was 55.2 years in physicians, 38.5 years in MAs and 46.3 years in stakeholders. In the second phase of data collection (2020), six additional in-depth interviews were conducted. Here, three interviews were conducted with experienced PCN management representatives. The remaining three interviews were carried out with physicians aiming at understanding the role of PCNs for primary care providers (*n* = 2) as well as gaining additional understanding about the computerized decision support system (CDSS), which was provided to practices in intervention arm C (*n* = 1). To support anonymity of the small sample of the additional in-depth interviews, socio-demographic characteristics are not reported here. The qualitative study collected data beyond the point of data saturation until the consistency of findings, as well as deviant observations, enabled assessment of data sufficiency. Table 2 describes the characteristics of the interview sample.

### 2.4. Implementation Program

#### 2.4.1. Uptake of Intervention Components

Figure 1 describes the uptake of study components across intervention groups (T2, *n* = 184). Since the different study components were rolled out at different times, the overall uptake was observed in T2 only. The highest uptake was reported for educational components addressed to healthcare professionals. Feedback reports, background information and the offered e-learning modules had the highest reported utilization rates, followed by disease-specific QCs. For QCs addressing respiratory tract infections, the highest rates of usage were reported. In contrast, the uptake of interdisciplinary QCs was heterogenous. Here, 57% of surveyed physicians noted to have used this format. The CDSS initially was offered to all 69 practices in intervention arm C. Since the technical integration took longer than expected, it could only be implemented for a shortened usage period in 34 practices (49.3%) where two particular types of administrative systems were in use.

Concerning patient information material, a difference in the usage of digital to analog components became apparent. Physicians across the three study arms reported high usage of informational patient flyers in German. Tablet devices were used by 33% of physicians in intervention arm B. In line with this, 30% of all surveyed physicians used the provided study-specific website, less than 10% of physicians noticed social media content addressing a rational use of antibiotics. A total of 47% of surveyed physicians reported that they had noticed the public campaign.

The perceived claiming of performance-based additional reimbursement (P4P) incentives differed noticeably across study groups. In intervention arms A and B, 64–66% reported that they took up the offer of additional reimbursement, compared to 37% in intervention arm C. A more detailed description of the uptake of intervention components across study groups is provided in Appendix A, Appendix A.

#### 2.4.2. Effort for Integration

From the MAs perspective, study components and newly gained knowledge were perceived to be smoothly integrable. Tendencies became obvious that analog study components were the easiest to be integrated into daily routines. Highest barriers were seen in integrating tablet devices where 50% of MAs anticipated high effort. Perspectives of MAs concerning the integration of study components are summarized in Table 3. (Appendix A provides further details referring to MAs perception).

#### 2.4.3. Perceived Reach and Impact

Physicians perceived to receive new impulses from intervention components. QCs, background information and feedback reports were reported to have the best ability in providing new understandings. This was followed by interdisciplinary QCs, which were seen as a positive influence on existing routines by more than half of the respondents. In the provided P4P, one third of respondents associated a provision of new impulses with this concept. The lowest influence was attributed to public campaign elements and the computerized decision support tool. Information about physicians’ perceptions regarding new impulses provided by intervention components is shown in Table 4. (More detailed information is provided in Appendix A.)

In the main interview study in 2018 (*n* = 45), physicians stated that participating in ARena had a major impact on their prescribing behavior and assumed this led to reduced antibiotic prescribing. This perception was also shared by physicians who considered themselves to have been low prescribers before study participation already. Physicians reconsidered the choice of antibiotics for more complicated infections and also reflected on an existing gap between guideline-recommendations and their previous prescribing behavior. ARena was seen as a constant reminder of a rational use of antibiotics. Hence, physicians stated they felt empowered in their choice of treatment in case uncertainties occurred. Another positive contribution was seen in a frequent participation in QCs. From their perspectives, QCs fostered a dialogue among physicians and helped to gain understanding about therapeutic decisions of other medical specialist groups. Physicians positively mentioned a perceived health literacy gain in patients. They experienced a decreased demand for antibiotics and observed sensitized patients who primarily aimed to avoid antibiotics. This led them to acknowledge that patient demand for ABs might be lower than initially expected.

One physician critically pointed to a potential selection bias in the ARena study by sampling healthcare providers who already had the intention to reduce their prescriptions. Other physicians mentioned that projects such as ARena should be repeated every few years to sustain effects and to provide information on current prescription rates and resistances.

“*So, you think more intensely about using one or the other antibiotic*.”A03#74

“*Patients became more sensitized as well and accepted reasons for holding back on antibiotics*.”Phys07#60

#### 2.4.4. Compatibility and Clarity

Major compatibility concerns were voiced with regard to offering tablet devices in waiting areas. A pediatrician stated that tablets contradicted his personal attitude and approach of a restricted use of digital media formats by children. Another reason for reluctance was a fear of being more and more replaced by digital applications. Others were willing to adopt the provision of tablets but observed a widespread patient disinterest. This was explained by the perception that patients primarily intended to come for personal consultation instead of receiving digitalized health-related information. Practical concerns were voiced in a general fear of theft of high value electronic devices and in hygiene issues, since physicians were reluctant to offer the devices to acutely infected patients.

Analog patient information material was reported to have a very high acceptance. Most interviewed healthcare providers wanted to sustain the utilization of flyers and posters beyond the study period. However, one physician considered flyers with rather playful designs challenging and contradicting the professional appearance when providing well educated patients with information that was to be taken for granted.

“*I’ve got to give something to take home. Something coming from me, reflecting my attitudes and this [flyer] didn’t suit me […]*.”Phys19#32

Physicians appreciated the concise information material provided in QCs and the feedback reports. They presumed that a more regionalized contemplation of antibiotic recommendations would be supportive. One physician pointed out that prescribing guideline-recommended antibiotics would not necessarily reflect current regional resistance situations. This physician suggested to develop tailored regional recommendations and constantly monitor and adapt them over time.

“*Of course, there has been the question which antibiotic works best in our region. So, there are differences which sometimes deviate from guideline recommendations*.”Phys11#40

### 2.5. Organizational Factors

#### Social, Political and Legal Factors

In order to sustain positively perceived effects of the ARena study, stakeholders as well as physicians reported on social and political aspects. Stakeholders assumed that sustainable change of antibiotics use could only be achieved by frequent public campaigns since behavior change requires time and repetition. Moreover, they suggested a mandatory yearly quality circle structure. Physicians complemented this by calling for more political support of network activities, since they felt that PCNs experienced little attention on the political stage and thus a disbalanced competition with medical care centers would be prevalent. Such care facilities managed by physicians of different medical specialties were seen as business entities which strongly pursued economic targets and minimization of financial risks and were less interested in care quality improvement. Furthermore, physicians pointed to a price gap between prescription drugs and over the counter drugs. Due to a lower financial patient contribution to prescription drugs, physicians reported to feel pressured to prescribe antibiotics to patients from low income households.

“*It would be good, of course, if these interventions which were quite accelerated, won’t be the last for the next ten years*.”NM#03#28

“*So, medical care centers with more than 80 employed physicians represent quite a market power and no primary care network can ultimately say: ‘Okay, we’re a conglomeration of established physicians, but in addition we are a power to be reckoned*.”Sh05#46

### 2.6. Incentives and Resources

Taking an economic perspective, physicians alluded to financial losses if consultation times constantly expanded. Therefore, they considered incentivization of documentation as well as consultation activities necessary to foster guideline-oriented prescribing sustainably. Using C-reactive protein testing was considered to provide safety for decision-making and thus physicians thought it should be reimbursed by health insurers. In reference to the P4P intervention component in ARena, its influence on decision-making processes was assessed to be heterogenous. While the survey data showed mixed results regarding the uptake of P4P reimbursements, interviewed physicians asserted it theoretically was the fastest way to change behavior.

“*If reimbursement changed, one question would be how to create incentives. Obviously, if reimbursement of ‘weaker’ medicine would be higher,—short consultation, writing a little prescription and off you go—that’s something basic, I suppose. So, less activism, less diagnostics, more talking, is not very well compensated, of course*.”Phys08#80

### 2.7. Primary Care Networks

Physicians considered participation in PCNs to be a motivational factor for guideline-oriented patient care and the implementation and adoption of new routines. In the T0 questionnaire, 70.5% of the physicians indicated that PCNs motivated guideline-oriented patient care. However, this assessment changed over time and decreased to 60% in T1. Awareness about offered training sessions regarding guideline-oriented antibiotics therapy stayed stable in the same period of time. Detailed perspectives of physicians on network participation is represented in Table 5.

In the interview study, it became apparent that the possibility of a continuous peer exchange was seen as a major advantage when joining a PCN. Physicians reported that they widely perceived a form of isolation in their small practices, which they could antagonize by their membership in a PCN. In addition, they stated that PCNs contributed to reducing professional insecurities, especially with regards to new treatment options. Also, physicians noted that PCNs supported their patients with a fast allocation of appointments with medical specialists in their network when necessary. Physicians attributed PCN membership to improved health services.

“*You have to educate yourself together, you have to know that others do it the same way because we’ve already seen clearly […] patients from external physicians come to us and I also do see it in the on-call practice, if patients come from external physicians, there is a difference between PCN physicians and non-PCN physicians in treatment procedures*.”Phys02#44

### 2.8. Individual Factors of Health Professionals and Patients

Health professionals positively acknowledged the educational flyers and information dissemination via a public campaign due to the desire of reliable information sources. Physicians considered socio-demographic patient characteristics to be of relevance regarding a successful implementation of digital health literacy interventions. They sensed the risk of excluding older patients by offering media channels they might feel overwhelmed and over-burdened with. Besides, they estimated health literacy competencies of younger adults to be more extensive than the provided information on the digital devices. Physicians also felt that a careful assessment of patient characteristics was necessary before delayed prescription strategies could be applied. Although these were perceived to meet high acceptance by patients, physicians reflected on cases where antibiotics were prescribed when they felt an ethical conflict because over-the-counter drugs could not be afforded by low-income patients.

Physicians acknowledged a decreasing patient demand for antibiotics. Nevertheless, they tried to meet patient preferences by intuitively applying behavior change techniques [11]. Although they were not part of the ARena implementation program, these approaches were used to educate patients about consequential harms of antibiotics and foster shared decision-making. Identified approaches were strategies of re-attribution, information about health consequences, pros and cons, information about social and environmental consequences, comparative imagining of future outcomes, credible sources and incompatible beliefs [11].

Physicians also acknowledged that they became aware of own misinterpretations regarding patient preferences through participating in ARena.

“*The desire for an antibiotic-based treatment noticeably decreased. The public campaign and flyers seem to have helped there as well. Patients are more frequently asking for complementary methods*.”Phys02#18

### 2.9. Capacity for Change

In T0, 90.2% of physicians and 76.6% of MAs reported to have implemented changes in practice in the last two years. A total of 50% of surveyed physicians further stated in T2 that the participation in ARena lead to a change in prescribing strategies. In the interview study, physicians reported that benchmarking procedures carried a high significance to them. Feedback reports would help to foster transparency of AB prescriptions and thus facilitate a decrease over time. Since the feedback reports provided during the ARena study required profound statistical knowledge, one physician additionally demanded explanatory meetings in which reports can be discussed with qualified peers. QCs were considered to be a suitable tool to receive latest information about the development of new antimicrobial resistances or the preferred choice of medication in pneumonia as well as urinary tract infections. Nevertheless, in order to guarantee a guideline-oriented treatment, they saw the requirement to restructure reimbursement schemes. Interventions such as counselling efforts, delayed prescribing or household remedies would need to be incentivized appropriately to sustainably guarantee a rational use of antibiotics.

“*Scientific investigations showed us that the provision of benchmark procedures alone gets the physician to prescribe less […] and yes, me personally, I consider this to be even more important than money*.”NM03#38

## 3. Discussion

This process evaluation was conducted alongside the ARena implementation program operated in a three-armed cluster randomized controlled trial design. A mix of methods was used to enable a sound understanding of working mechanisms, contextual factors and personal beliefs of participants influencing the uptake and dissemination of study components. The longitudinal design of the survey study provided insights into participants’ changes in beliefs about components and appropriate use of antibiotics over the time of evaluation. Based on the overarching explanatory framework, insights from the interview study supported understanding of survey findings and vice versa regarding the uptake of intervention components and potential influences on prescribing routines.

Fostering appropriate antibiotic use for the treatment of non-complicated infections by addressing primary care physician, their team and patient knowledge and attitudes about the use of antibiotics is a complex stewardship endeavor in a dynamic field where interventions need to be integrable into daily practice with little effort, yet still need to carry the potential to reach the desired impact. Ideally, such intervention components require a relatively small commitment of resources on the healthcare provider side, increase guideline concordance and foster a decrease of overuse and inappropriate prescribing [12]. Different models of implementing such antimicrobial stewardship programs in primary care have already been evaluated, including physician education [13,14], audit and feedback [15,16,17], electronic clinical decision support [18], peer comparison [19,20,21], and more [22,23]. To our knowledge, this process evaluation is the first to evaluate an implementation program that combined several of these components and tested them in a primary care network setting. Thus, it adds to the growing evidence base informing programs which aim to support physicians in primary care to reduce inappropriate prescribing. Following, key findings of the combined data will be discussed and a comparison to prior work is drawn addressing each of the analyzed study component.

The highest utilization rates of study components were observed in professional educational audit and feedback study components, namely in feedback reports, evidence-based background information and QCs. Healthcare professionals considered these components to be most beneficial to changing own prescribing routines. In particular, the interactive QCs encouraged physicians and MAs to reflect on own routines and attitudes by providing current evidence-based information to initiate change in existing routines. Benefits of benchmarking procedures were emphasized as well. Analog study components addressing patient education were perceived as easily integrable into existing routines. Contrarily, providers saw challenges for the integration of the tablets into workflows. The offered website and social media content were geared to provide reliable information to patients, not the physicians, and thus there was little engagement on their side. The CDSS; however, was considered helpful with integrating knowledge into daily routines and choosing indication-appropriate antibiotics.

Interviewed physicians saw P4P strategies as a key to generate behavior change. In the survey; however, only half of the participating physicians reported to have claimed the reimbursement. One explanation could be that such a financial incentive encourages participation in a study but is of lesser importance after this decision. Overall, a vision of recurring implementation programs similar to ARena was emphasized and regular thematic updates were considered necessary to sustain effects beyond the study period.

### 3.1. Comparison to Prior Research

Audit and feedback are considered to be intervention components which improve health professionals’ compliance with desired practice. Regarding QCs and feedback reports as audit and feedback procedures aiming at optimizing professional practice and healthcare outcomes, a systematic review identified determinants to increase effects of this approach: Procedures were most effective when initial baseline performances of respective units were low, the information source was a supervisor or a peer, audit and feedback were provided more than once, delivered in written and verbal formats, and targets as well as action plans were included [24]. Particularly, the determinant to receive feedback from colleagues is mirrored in our data. Since antibiotic prescribing rates have been decreasing in German primary care in recent years [3,25,26,27,28] and the observed participant prescribing rates for acute non-complicated infections were moderate prior to the ARena project [29], outcome analyses based on the claims data will inform about the extent of effects that were possible to be achieved by ARena, particularly regarding the guideline-conform use of recommended antibiotics.

The appreciation of benchmarking procedures identified from the qualitative data in ARena has also been found to be effective in prior research [19]. Aiming to foster appropriate antibiotic prescribing for respiratory tract infections during ambulatory visits, the authors identified a decrease of inappropriate prescription rates from 19.9% to 3.7% in 18 months attributable to peer comparison mechanisms. Such peer mechanisms and social influences have been identified in ARena as well [30].

Interviewed stakeholders suggested frequent public campaigns and QCs to ensure sustainable effects. The idea of repetitive campaigns is supported by a systematic review on public campaigns addressing antibiotic use in outpatient settings in high income countries [31]. The review found public campaigns to be most effective if designs were multifaceted and repeated over several years, carried clear and simple messages and avoided threats. Previous investigations further identified social media platforms to be a preferred channel of the public to find antibiotic-related information. In Italy, 46.5% of social media users consulted respective platforms as information source [32]. The healthcare professionals in ARena were not directly addressed by the delivered social media content and perspectives of the public could not be evaluated. Future projects comprising educational social media campaigns could therefore design target group-oriented dissemination strategies and evaluation concepts to achieve a wider reach and understand potential effects.

Interviewed physicians pointed to a potential discrimination against elderly patients by confronting them with digital educational solutions, rated health literacy competencies of the younger population to be higher and saw employed patients at a higher risk of demanding antibiotics due to work-related stressors. These perceptions are supported by prior research [33]. A connection between age and antibiotic-related health literacy has not been identified. However, people who had been taking antibiotics over the last 12 months seemed to be better informed than those who did not take any antibiotics [34]. Regarding the elderly’s ability to engage in e-health solutions, results of a cross-sectional survey indicate a potential exclusion of this patient group since individuals showed less access and experience with digital devices [35]. Taking this into account, current approaches targeting health literacy competencies may need to be more specifically adapted to the needs of the respective patient group.

The hesitation of healthcare professionals towards tablet devices as one source of patient information has been reported in a recent study in Germany where general practitioners and MAs were found to be reluctant to provide tablet devices in the primary care setting [36]. Fear of theft, hygiene issues and being contradictory to physician’s personal values were identified as main challenges. Further determinants inhibiting a successful integration of tablet devices were also identified by recent research and comprised poor digital health literacy, added workload, lack of motivation and a miss-fit to organizational structures [37].

With regard to the CDSS applied in Arena, which was built into the practice administrative software system, users appreciated to have guidance in prescribing decisions. This is in line with previous findings which investigated the acceptance of decision support systems in psychiatrists [38]. Based on an online survey conducted in Germany in 2019 [39], the main reasons for inadequate antibiotic prescribing by German physicians were a lack of knowledge and a limited data availability about regional resistance situations. Therefore, it was aimed to provide necessary data via CDSS. Nevertheless, such systems have not yet been proven to be effective since content, designs and evaluation strategies across studies are heterogenous and evidence about CDSS addressing antibiotic prescribing in ambulatory care is rather low [40]. Once analysis of the primary and secondary outcomes of the ARena study based on claims data will be concluded, indications of a potential effectiveness of the applied CDSS as part of a bundle of intervention components may contribute to closing this gap. However, a recent study in Germany also found that the affinity for interaction with technology varied widely among GP trainees with an average age of around 35 and concluded that a better understanding of such systems and more support in learning to use specific features was necessary, since relevance of information technology systems in healthcare is high and potentially will increase in coming years [41].

Interviewed physicians saw P4P strategies as a key element for behavior change, yet the uptake in ARena did not match this perception. A recent survey study conducted among family physicians to understand awareness and attitudes towards participation in P4P programs identified major reasons for rejecting additional reimbursements in increased loads of administrative work (79.6%) and inadequate understandings of the P4P content (62.9%) [42]. This cannot be supported by our data but could explain the heterogenous results.

On system level, interviewed stakeholders and physicians suggested regional antibiotic prescribing recommendations and reimbursement of point-of-care testing by statutory health insurances. In terms of regionally tailored recommendations, the German Antibiotic Resistance Strategy (DART 2020) [43] defined the goal to detect resistance developments at an early stage and thus national and European surveillance systems noticeably expanded [6]. Findings of the process evaluation in ARena indicate that a more effective communication between surveillance researchers and healthcare providers should be fostered. In terms of cost absorption of point-of-care testing, research found moderate evidence that using CRP tests supports a decrease in antibiotic-prescribing [40], so it may be justified to further discuss this option.

Though knowledge dissemination about behavior change techniques was not part of the ARena implementation program, physicians reported their intuitive use to antagonize patients’ antibiotic-demands. Such intuitive use of behavior change techniques could be supported by applying the Tailored Antimicrobial Resistance Program (TAP), which was developed by the World Health Organization (WHO) [44]). TAP as a concept targets the conversion of habits supporting antimicrobial resistances by offering guidance in design, implementation and evaluation of such strategies. It was tested in a pilot study designed in a stratified cluster randomized trial. During an implementation period of eight weeks, patient requests for antibiotics significantly decreased from 60.2% to 38.5% (*p* < 0.05) [44]. This strongly suggests that using behavior change techniques might be a powerful tool in the context of antibiotic resistances and indicates that future interventions could focus them more closely.

PCNs can shape healthcare by driving optimized performance and innovation, notably when effective network strengths meet new approaches to healthcare [45]. Such physician peer networks may be seen as a potentially important factor that might influence medical practice [46]. Being part of a PCN presumably supports adoption of specific behaviors. This might be attributable to social contagion as an influencing process where the network members are impacted by each other in adoption decisions [47]. Social contagion theory suggests that human behaviors and traits can spread in social networks [48] and assumes this is fostered by behavioral mechanisms of imitation, role modelling and persuasion [49]. Physicians who interact in networks are likely to share beliefs and experiences with each other and their practice patterns may be influenced by such interpersonal information exchange [50] and activate peer influence as a potential driver of physicians’ practice styles [51]. Our findings indicate that the PCNs as a setting strengthened the physicians and fostered necessary changes by taking the responsibility for change from the individual to the collective setting. This, combined with the audit and feedback component and the interactive QCs, made the value of the study very tangible for participants.

### 3.2. Strengths and Limitations

In this process evaluation, measures were based on a widely tested conceptual framework. A mix of methods guaranteed a triangulation of data and a sound sample size performing high response rates in both qualitative and quantitative research components assured data saturation to a satisfactory degree. The total number of conducted interviews closely follows the study protocol [8] and exceeds the targeted sample size by one participant. The longitudinal survey design helped to detect changes over three measuring points and a combination of an a priori as well as a de novo approach of interview data strengthened the analysis. Qualitative data analysis followed standardized procedures of COREQ guidelines [52].

Quantitative analysis was exclusively reported descriptively and does not allow any prognostic conclusions. Since the analysis of claims data regarding primary outcomes of ARena is not yet completed, findings of the process evaluation cannot be contextualized in relation to the primary outcomes. The patient perspective was not considered within this process evaluation but, rather, evaluated separately [53]. The study setting in PCNs may have contributed to amplified effects which should be considered when transferring findings to routine care. A potential selection bias may be present in pre-existing participant motivation to reduce antibiotic use. It is possible that socially desirable answers were provided. Participant age and significant work experience potentially limits the transferability of findings to a younger generation of medical professionals.

## 4. Materials and Methods

### 4.1. Study Design

ARena was conducted as a three-armed cluster randomized trial and implemented in the primary care setting of 14 PCNs in two German federal states (Bavaria and North Rhine-Westphalia). In arm A (4 PCNs), components included a standard set comprising e-learning on communication, moderated QCs and data-based feedback for physicians, public information campaigns, P4P and printed culture-sensitive information material for patients. During the intervention period, all participating PCNS could attend QCs at four different times. These were offered to foster critical discussion and assessment of clinical practice and focused on key issues related to care quality and appropriate use of antibiotics regarding respiratory tract infections, urinary tract infections, community acquired pneumonia and multi-resistant pathogens. In addition to this standard set, arm B (5 PCNs) addressed MAs with an e-learning module on communication and separate QCs for MAs only. Additionally, tablet pcs providing patient information material were offered to be used in waiting areas. In addition to the standard set, arm C (5 PCNs) received a CDSS and multidisciplinary QCs. The CDSS was embedded in two commonly used administrative practice software systems with the aim of supporting prescribing decisions and were only offered in arm C. Standard care was reflected by an added cohort based on claims-data. The study protocol [8] provides a detailed description of ARena and its’ interventions.

The process evaluation used a mixed-methods approach. All physicians included in the intervention and MAs in intervention arm B received study-specific questionnaires at three different points in time. In addition, open-ended semi-structured interviews were conducted with primary care physicians (general practitioners, otorhinolaryngologists, urologists and pediatricians), MAs and stakeholder representatives (health insurance and patient representatives, and PCNs management). Interview data were supplemented by a one-time socio-demographic survey of interviewees. A different interview guide was developed for each participant group based on the pre-defined research questions and a literature review.

### 4.2. Study Population for Survey

Survey questionnaires (T0, T1, T2) were sent out at three different points in time during the study to all physicians included in ARena (T0 *n* = 303, T1 *n* = 312, T2 *n* = 292) and MAs employed at participating practices allocated to intervention arm B (T0 *n* = 84, T1 *n* = 88, T2 *n* = 85). Due to fluctuation, participant numbers varied. To increase response rates, e-mail reminders were sent after 4 weeks each time. PCN managers (*n* = 14) were additionally invited to participate in an online questionnaire reflecting the role of PCNs in the ARena project.

### 4.3. Study Population for Interviews

A total of 120 physicians (40 in each of the three intervention groups with even distribution regarding gender and PCNs) were randomly selected and invited for participation in the interview study by e-mail via the aQua Institute, Goettingen. An e-mail reminder was sent after three weeks. In intervention arm B, 25 MAs were invited for participation. Previous experiences and anticipated response rates were the basis for calculation of the number of contacted potential recruits and it was aimed to approximate the targeted number of interviews as defined by the study protocol [8]. The ARena study team at the Department of General Practice and Health Services Research, University Hospital Heidelberg (DGPHSR-UH-HD), recruited a sample of 45 participants between March and May 2018, using a purposive strategy which aimed at even distribution regarding gender and intervention groups. Potential recruits for the process evaluation were all physicians who participated in ARena, MAs who participated in intervention arm B as well as managerial stakeholder representatives of participating PCNs, health insurance providers, association of statutory health insurance physicians and of a self-help organization. Participants had to be >18 years old, legally fully competent and fluent in German. All interested parties meeting all inclusion criteria received printed and verbal information detailing the process evaluation (via postal mail and phone call) and had to return a signed written informed consent form prior to the interview.

All of the stakeholder representatives (*n* = 7) were known contacts and were personally addressed by the aQua Institute via e-mail. They were sent a personalized cover letter, a detailed description of the study and the process evaluation and a feedback form to be returned by fax or e-mail to state interest in participation.

### 4.4. Data Collection and Analysis

#### 4.4.1. Survey Study

Study-specific questionnaires were sent by mail in January 2018 (T0), October 2018 (T1) and July 2019 (T2). The survey focused on the provided intervention components, relevant context factors, prescribing decisions and general perceptions regarding antibiotics. In addition, questionnaires T1 and T2 itemized interim and concluding assessments of intervention components. Between February and April 2018 (T0), November 2018 to January 2019 (T1), and July to September 2019 (T2), all completed questionnaires were received and registered by the study team at the DGPHSR-UH-HD. Data from all returned questionnaires were digitalized, transferred into IBM SPSS Statistics 24 (IBM, Armonk, NY, USA). and analyzed descriptively. Only answered survey items were included, missing values were marked by a specific code.

In March 2020, the online survey addressed PCN management representatives to further explore characteristics of the participating networks and uptake and implementation of the program from their perspectives. Findings from all survey data are reported here.

#### 4.4.2. Interview Study

Between April and June 2018, interviews with participating physicians (*n* = 27) were carried out and digitally audio recorded by three researchers (RPD, MK, AS) of the study team at the DGPHSR-UH-HD. A semi-structured interview guide was used to gain insight into typical practice regarding antibiotic prescribing, consideration of patient preferences, implications of intervention components for patient care, general contextual factors and the role of the PCNs. In April and May 2018, all interviews with MAs (*n* = 11) and stakeholders (*n* = 7) were conducted and audio recorded by two researchers (RPD, MK). The MA interview guide focused on their perspectives and experiences. The stakeholder interview guide covered expectations of a potential impact of intervention components, perspectives on contextual factors, and recommendations for the future use of antibiotics. During April and Mai 2020, one researcher (RPD) conducted and audio recorded additional in-depth interviews (*n* = 6) with PCN managements and participating physicians to further explore aspects regarding the role of the PCNs within the project and sustainability of perceived intervention component effects. Translated interview guides are provided in Appendix A: Interview guide physicians (translated), Appendix A: Interview guide medical assistants (translated) and Appendix A: Interview guide stakeholders (translated). Table 6 outlines the data collection sources for findings presented here.

All interviews (*n* = 51) were conducted over telephone. For the analysis of the pseudonymized verbatim transcripts, a thematic framework analysis [54] based on the Tailored Implementation for Chronic Disease (TICD) framework was used. The TICD offers seven domains to classify determinants of implementation (‘Guideline factors’, ‘Individual health professional factors’, ‘Patient factors’, ‘Professional interactions’, ‘Incentives and resources’, ‘Capacity for organizational change’ and ‘Social, political and legal factors’) [55]. In accordance with the ARena study protocol [8], selected pre-defined TICD categories were applied to identify determinants of practice with regards to the appropriate use of antibiotics in acute non-complicated infections and potential changes in health professional practice in primary care. A priori, themes of interest were identified deductively from the TICD framework in three key categories: ‘guideline factors’ (re-coded to ‘implementation program’), ‘organizational factors’ and ‘individual factors’. The subcategory ‘primary care networks’ was identified inductively de novo from the data itself by the interprofessional team of researchers (Public Health and Health Services Research) and assigned to ‘organizational factors’. Figure 2 provides a comprehensive display of the analytical approach. Two researchers (RPD, MK (*n* = 45) and RPD and LK (*n* = 6)) coded the transcripts iteratively and independently in MAXQDA Analytics PRO 18 (Versions 18.0.3 and 18.2, VerbiSoftware, Berlin, Germany). With the aim to ensure a wide consensus and intercoder congruity, divergent codings were discussed. Descriptive analysis of participants’ socio-demographic characteristics was performed using IBM SPSS Statistics Version 24.

### 4.5. Ethics Approval and Consent to Participate

This study received ethical approval by the medical ethics committee of the Medical Faculty of Heidelberg University (S-353/2017). Participants in the process evaluation all gave written informed consent. Confidentiality and anonymity were ensured throughout the study.

### 4.6. Availability of Data and Materials

All data generated and analyzed during this study are stored on a secure server at the DGPHSR-UH-HD. De-identified sets of the data collected and analyzed during this study can be made available by the corresponding author on reasonable request.

## 5. Conclusions

This process evaluation identified individual and organizational factors affecting the feasibility of this multi-faceted program and provided indications regarding the potential implementation of tested components into routine care. Though approaches targeting health literacy competencies and clinician’s therapy decisions at the same time may need to be specifically tailored to the needs of respective targeted groups, audit and feedback reports in combination with evidence-based information provided and discussed in QCs should be established in primary routine care to reduce the overuse of antibiotics.

## Figures and Tables

**Figure 1 antibiotics-09-00878-f001:**
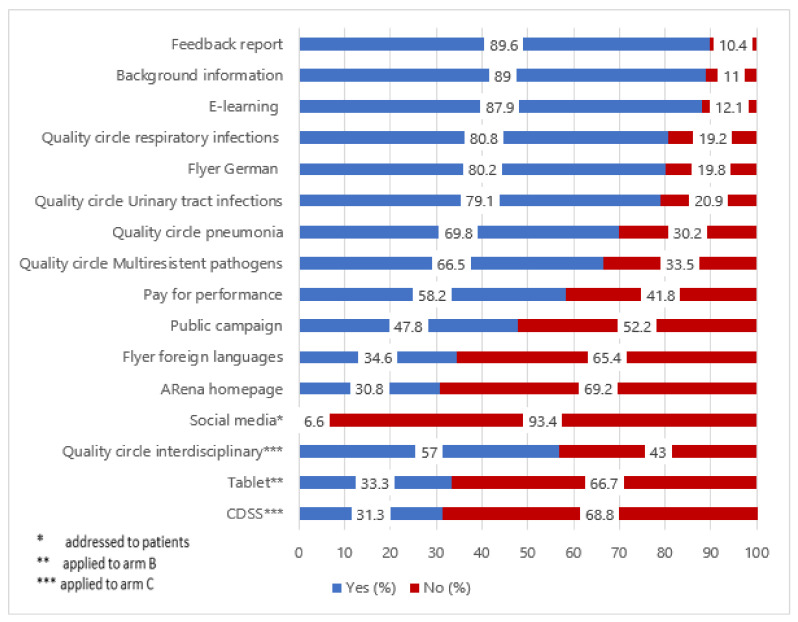
Uptake of intervention components (T2, *n* = 184).

**Figure 2 antibiotics-09-00878-f002:**
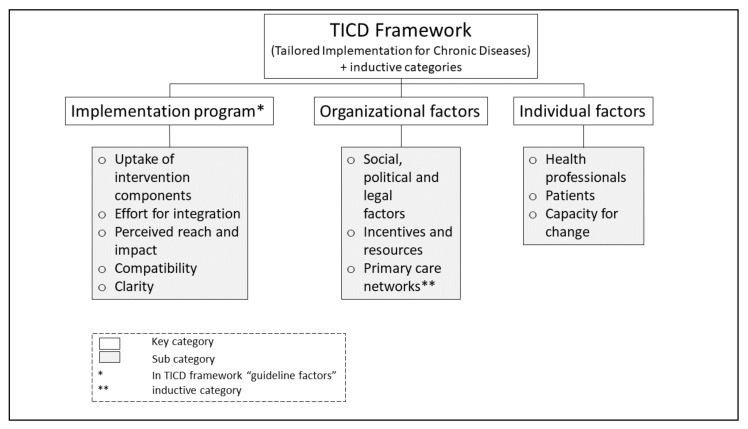
Analytical approach based on the Tailored Implementation for Chronic Disease (TICD) framework.

**Table 1 antibiotics-09-00878-t001:** Characteristics of survey respondents for measuring points T0 and T2.

Survey Respondents (T0)	*N*	Physicians	Medical Assistants	Total
Sex (f/m) *n* (%)	304	76/148 (34.0/66.0)	80/0 (100/0)	156/148 (51.3/48.7)
Age in years (range) (mean)	299	35–73 (54.4)	19–61 (38.7)	19–73 (46.5)
Years of working experience (range) (mean)	306	5–48 (25.4)	1–40 (19.2)	1–48 (22.3)
Working in general practice (%)	309	75.3	76	75.6
medical practitioner in private practice years (range) (mean)	220	1–41 (17.7)	N/A	220 (17.7)
Network member years (range) (mean)	207	0–28 (10)	N/A	10
Participating in network events times/year (range)	217	7.3 (0–50)	N/A	7.3 (0–50)
**Survey Respondents (T2)**				
Sex (f/m) *n* (%)	240	59/125 (32/68)	56/0 (100/0)	115/125 (48/52)
Age in years (range) (mean)		35–73 (54.2)	19–61 (39.5)	19–73 (46.9)
Experience in years (SD) (mean)		7.9 (26.4)	12.9 (19.3)	24.8 (9.8)

N/A = Not applicable.

**Table 2 antibiotics-09-00878-t002:** Characteristics of the interview sample.

Interview Participants (in 2018)	*N*	Physicians	Medical Assistants	Stakeholder	Total
Gender f/m (%)	45	9/18 (33/66)	11/0 (100/0)	3/4 (43/57)	23/22 (59/41)
Age in years range (mean)	45	43–66 (55.2)	20–60 (38.5)	31–63 (46.3)	31.3–63 (46.6)
Years of experience in current position range (mean)	45	10–38 (26)	2–40 (19)	1–10 (5.8)	1–40 (17)
Working in general practice (%)	38	66.6	81.8	N/A	74.2
Employed part-time *n* (%)	4	1 (2.7)	3 (27.3)	N/A	4 (8.88)
PCN * member in years range (mean)	27	2–23 (10)	N/A	N/A	10
Additional qualifications *n*	7	N/A	7	N/A	7
**Interview participants (additional; in 2020)**					
Sex f/m (%)	6	2/1 (66/33)	N/A	1/2 (33/66)	3/3 (50/50)
Age years range (mean)	6	58–66 (60.7)	N/A	44–55 (49.7)	44–66 (55.1)
PCN * management function years range (mean)	3	N/A	N/A	8–22 (13)	8–22 (13)
PCN * member in years range (mean)	2	9–22 (15.5)	N/A	N/A	9–22 (15.5)

* PCN = Primary care network. N/A = Not applicable.

**Table 3 antibiotics-09-00878-t003:** Medical assistant (MA) perspective on integrating study components and newly gained knowledge (T2; *n* = 58).

Integrating Study Components into Practice Routines Was Associated with Great Effort	Agree * *n* (%)
Flyer German	4 (6.9)
Flyer foreign languages	6 (10.3)
Website	14 (24.1)
Public campaign	17 (29.3)
Social media content	20 (34.5)
Tablet device	29 (50.0)
**Transferring Newly Gained Knowledge Was Associated with Great Effort**	**Agree *** ***n* (%)**
Content of online training	9 (15.5)
Content of background information	11 (19.0)
Content of quality circles	13 (22.4)
Content of feedback reports	13 (22.4)

* Consolidation of five-point Likert scale values “Strongly Agree” and “Agree”.

**Table 4 antibiotics-09-00878-t004:** Physician perspective on new impulses provided by study components (T2, *n* = 184).

The Intervention Component Provided New Impulses	Agree *(%)		
Intervention Arm	A	B	C
Online training	48.5	59.7	40.8
Quality circles	72.1	83.6	79.6
Feedback report	58.8	65.7	55.1
Background information	73.5	68.7	73.5
Patient flyer German	35.3	53.7	53.1
Patient flyer foreign	13.2	19.4	22.4
Website	14.7	14.9	20.4
Social media	5.9	4.5	6.1
Public campaign	26.5	26.9	36.7
Pay for performance	32.4	37.3	30.6
Tablet device	N/A	9.0	N/A
Interdisciplinary quality circles	N/A	N/A	53.1
Decision support tool	N/A	N/A	16.3

* Consolidation of five-point Likert scale values “Strongly Agree” and “Agree. N/A = Not applicable.

**Table 5 antibiotics-09-00878-t005:** Physician perspective on network participation (T0 *n* = 229; T1 *n* = 200).

Participating in the Network	Agree T0/T1 (%)	Neutral T0/T1 (%)	Disagree T0/T1 (%)
provides motivation for guideline-oriented patient care	70.5/60.0	18.5/19.0	11/22.0
furthers shared-decision making	60.8/59.0	19.8/24.0	19.4/18.0
supports management of patient expectations	61.2/51.0	21.6/30.0	17.2/19.0
supports implementation of new routines	74.0/59.0	16.3/18.0	9.7/24.0
impacts antibiotic prescribing decisions	43.3/36.0	22.1/22.0	34.5/43.0
**In My Primary Care Network**	**Agree** **T0/T1 (%)**	**Neutral** **T0/T1 (%)**	**Disagree** **T0/T1 (%)**
antibiotics therapy is discussed	89.5/86.0	8.8/10.0	1.7/4.0
peer exchange on guideline-oriented antibiotics therapy is facilitated	79.9/79.0	14.5/11.0	5.6/10.0
exchange about antibiotic prescribing routines for non-complicated infections is contingent	71.5/73.0	18.4/16.0	10.1/11.0
there are conventions about the use of antibiotics for non-complicated infections	65.8/72.0	21.5/16.0	12.7/12.0
training on guideline-oriented antibiotics therapy is offered	89.0/75.0	6.6/18.0	4.4/7.0
I have taken part in training on guideline-oriented antibiotics therapy	89.0/87.0	6.6/9.0	4.4/4.0

**Table 6 antibiotics-09-00878-t006:** Data collection sources and numbers of participants.

Source	Physicians	Medical Assistants	Stakeholders	Description
Interviews (n)	27	11	7	Over telephone
Socio-demographic questionnaire (n)	27	11	7	Paper based
Thematic in-depth interviews (n)	3	N/A	3	over telephone
Survey T0 (n)	229	80	N/A	Paper based
Survey T1 (n)	200	73	N/A	Paper based
Survey T2 (n)	184	58	N/A	Paper based
Online survey (n)	N/A	N/A	10	Online

N/A = Not applicable.

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
