# Peer review of "Fostering Appropriate Antibiotic Use in a Complex Intervention: Mixed-Methods Process Evaluation Alongside the Cluster-Randomized Trial ARena"

_antibiotics, 2020, doi:10.3390/antibiotics9120878_

Round 1

Reviewer 1 Report

Antibiotic stewardship in healthcare facilities is an important topic and this manuscript attempts to draw some conclusion and evidence-based recommendations by analysis of data derived from a 2017-2020 German epidemiological study. While the topic and observational data are of great importance and merit, at the current state, the study has major inconsistencies in style and presentation of the data. The study appears to have very small sample size for some of the analysis that makes the current results and recommendations only exploratory in nature and perhaps not generalizable. I would need to sadly recommend rejecting the paper at current state and encourage a resubmission after analysis with larger data set, meeting the journal guidelines, and major improvements in graphical and tabular presentations of the data. Here are some additional comments:

-Introduction of the study is very brief and missing important and critical references. For example, in addition to the report cited from WHO, information about most recent report of CDC (an perhaps equivalent information form ECDC) could be further discussed.

CDC Antimicrobial resistance report: https://www.cdc.gov/drugresistance/pdf/threats-report/2019-ar-threats-report-508.pdf 

The cited WHO reference also requires correction by providing the weblink to the report that was used for the study.

-This manuscript is not following the authors guideline of the study, there are different font size thought the manuscript and headings and subheading are not in harmony with the guidelines. Just as an example, on Line 222 all words are initiated with capitalized letters or on line 423 there is a heading that is unclear if it is a heading to actually a subheading to discussions.

-Graphical and tabular presentations of the study additionally require considerable amount of work and improvement. Table 1 as an example has empty spaces between the rows and various color are used for lines. Table three has also major formatting issues, as an example “r” in “stakeholder” are extended to a new line that makes the table visually flawed. The table also would need to have sufficient information in the heading and/or as sub-note to assure better assimilation of the content by antibiotics readership. For example, authors could explain what “Phys,” “MA,” and “T0,” and “T2” are in the table description and/or food note. It is unclear why authors are not discussing T1 responses in this table.

-Most importantly the current study has such a small sample size for some of the analyses that the reviewer has concern about external validity of the results and generalizability of the outcome. Based on information from table 6, there were only 7 stakeholders involved in the study and only 3 of them were part of the study for “Thematic in-depth interviews.”

Author Response

Plese see the attachment for our point-by-point response.

Reviewer 2 Report

Antibiotic resistance and misuse are of major importance to public health and the authors have advanced a very well-written and designed study addressing this problem in Germany, but which has global implications. 

Overall, this is a very strong paper and I wish to congratulate the authors on their efforts. I particularly liked the idea of adding selected quotes from study participants as it brings the paper to 'life".

Author Response

Thank you very much Reviewer 2 for taking the time to assess our manuscript and for your kind words. 

Reviewer 3 Report

Thanks for this study. As the majority of antibiotics are prescribed in the community setting, research into the acceptance of targeted stewardship interventions is pertinent. You performed a mixed design (quantitative-qualitative) study in a cohort of healthcare providers who consented to participate in an AMS intervention.

  1. I would suggest to move the methods section after the introduction.
  2. Table 1: could you enlighten what you exactly mean with 'resident years'; i.e. in living in Germany?
  3. Section 'strengths and limitations': can you identify any sources of bias in your study, to what extent could the results be generalized to other populations? 

Author Response

Please see the attachment for our point-by-point answers to your comments.

Reviewer 4 Report

The work presented in the manuscript is part of a large project, the ARena program, focusing on the perception of different actors about the implemented program. It is a very complex and interesting work, however, it is important that the authors clarify some points in order to facilitate the understanding and the reading of the potential readers.

The objective of the work here presented, is not well defined. Please clearly specify the purpose of the work presented, in the abstract and in the end of the introduction.

The methods section should explain more detailed how participants were selected and recruited for the study presented in this manuscript. All the participants were exposed to all the interventions during the implementation of the ARena program.

In lines 186-187 “The lowest influence was attributed to public campaign elements and the computerized decision support tool”. However, no data were reported in what concerns the computerized decision support tool, by A and B groups. “N/A” was reported by the participants or this intervention was not applied in the groups' A and B? It is important to clarify in the methods section how was the exposition to the support tools. Had the computerized systems integrated into the prescription system? Or were delivered as an app for physicians personal use?

It would be interesting to discuss the data obtained regarding the perception of different stakeholders about the interventions performed, with the impact that these interventions had on the prescription and use of antibiotics.

Author Response

Please see the attachment for our point-by-point answers to your comments and sggestions. 

Round 2

Reviewer 1 Report

Authors have improved their manuscript to great extent and are stablished researchers in their fields. But sadly this particular manuscript in my opinion does not have generalizable outcome due to very small sample size. I understand the authors discuss feasibility and funding consideration for larger trials, but this study is simply under-powered, and results might not be generalizable. In the revised manuscript, authors discuss this major limitation of their studies in details by adding new sections. But this limitation could be incorporated in the tile and abstract as well to provide a fair summary of the study for the readers and antibiotics stakeholders. I would suggest adding a brief section to abstract and verbiage “ preliminary study of” to title.
